# Study protocol: an investigation of the prevalence of autism among adults admitted to acute mental health wards: a cross-sectional pilot study

Sam Tromans [1,2] Guiqing Lily Yao,[1] Reza Kiani,[1,2] Regi Alexander,[1,3] Mohammed Al-Uzri,[1,4] Traolach Brugha[1,4]

¹Department of Health Sciences, University of Leicester, Leicester, UK
²Learning Disability Psychiatry, Leicestershire Partnership NHS Trust, Leicester, UK
³Learning Disability Psychiatry, Hertfordshire Partnership University NHS Foundation Trust, Norwich, UK
⁴General Adult Psychiatry, Leicestershire Partnership NHS Trust, Leicester, UK

**Correspondence to**
Dr Sam Tromans;
sjt56@leicester.ac.uk

## ABSTRACT

**Introduction** Autism spectrum disorders (ASDs) are associated with difficulties in social interaction, communication and restricted, repetitive behaviours. Much is known about their community prevalence among adults, data on adult inpatients within an acute mental health setting is lacking.

This pilot study aimed to estimate the prevalence of ASDs among adults admitted to acute mental health wards and to examine the association between ASDs and psychiatric and physical comorbidities within this group.

**Methods and analysis** A multiple-phase approach will be used. Phase I will involve testing of 200 patients and corresponding informants, using the autism quotient (AQ), the informant version of the Social Responsiveness Scale, second edition—Adult, the self and informant versions of the Adult Social Behaviour Questionnaire and the EuroQol-5D-5L. Patients with intellectual disability (ID) will bypass Phase I.

Phase II will involve diagnostic testing of a subgroup of 40 patients with the Diagnostic Interview for Social and Communication Disorders, the Autism Diagnostic Observation Schedule version 2 and the ASD interview within the Schedules for Clinical Assessment in Neuropsychiatry version 3. 25±5 patients will not have ID and be selected via stratified random sampling according to AQ score; 15±5 patients will have ID. Phase II patients will be interviewed with the Physical Health Conditions and Mental Illness Diagnoses and Treatment sections of the 2014 Adult Psychiatric Morbidity Survey.

Prevalence estimates will be based on the proportion of Phase II participants who satisfy the 10th revision of the International Statistical Classification of Diseases and Related Health Problems Diagnostic Criteria for Research (ICD-10-DCR) and the 5th edition of the Diagnostic and Statistical Manual of Mental Disorders (DSM-5) diagnostic criteria for ASD, adjusting for selection and non-response. Univariate analysis will be conducted for comorbidities to identify the level of their association with an ASD diagnosis.

**Ethics and dissemination** Study oversight is provided by the University of Leicester. The National Health Service Health Research Authority have provided written approval. Study results will be disseminated via conference presentations and peer-reviewed publications.

**Trial registration number** ISRCTN27739943

### Strengths and limitations of this study

► The study is currently limited to a single site, limiting its generalisability. However, applications to extend to other sites are currently in progress.
► The planned sample size is in keeping with a pilot study, although there are plans to expand to a larger trial on completion.
► The primary outcome measure (meeting the diagnostic criteria for autism spectrum disorders (ASDs)) is somewhat contingent on the diagnostic criteria used, although internationally recognised criteria have been used (the ICD-10-DCR and DSM-5).
► The study will be conducted under double-blind conditions, whereby neither patients nor health professionals conducting Phase II clinical assessments will have knowledge of the participants' corresponding Phase I questionnaire results.
► The study design allows consideration of patients with and without intellectual disabilities separately, in contrast to many other studies of its type.

## INTRODUCTION

Autism spectrum disorders (ASDs) are life-long neurodevelopmental conditions characterised by impairments in reciprocal social interaction and communication, as well as restricted, stereotyped and repetitive behaviours.[1] These difficulties should manifest prior to 3 years of age, thus a detailed developmental history forms a key part of the diagnostic assessment process. Although originating in childhood, ASDs follow a persistent course throughout adult life.[2] While the level of impairment associated with ASDs varies between individuals, the impact on both those with ASD and their families is always profound.[3]

The diagnostic criteria for ASD according to the ICD-10-DCR[4] and DSM-5[5] diagnostic classification systems are summarised in box 1. It should be noted that the transition from DSM-IV to DSM-5 criteria for

> **Box 1  Summaries of the ICD-10-DCR4 and DSM-55 diagnostic criteria for ASD. For the diagnostic criteria in full, please refer to the source texts.**
>
> **ICD-10-DCR Criteria for Childhood Autism\***
> A. Presence of abnormal or impaired development before the age of three years
> B. Qualitative abnormalities in reciprocal social interaction
> C. Qualitative abnormalities in communication
> D. Restricted, repetitive and stereotyped patterns of behaviour, interests and activities
> E. The clinical picture is not attributable to the other varieties of pervasive developmental disorder; specific developmental disorder of receptive language with secondary socioemotional problems; reactive attachment disorder or disinhibited attachment disorder; mental retardation with some associated emotional or behavioural disorder; schizophrenia of unusually early onset; and Rett's syndrome.
>
> **DSM-5 Criteria for Autism Spectrum Disorder**
> A. Persistent deficits in social communication and social interaction across multiple contexts
> B. Restricted, repetitive patterns of behaviour, interests or activities
> C. Symptoms must be present in the early developmental period
> D. Symptoms cause clinically significant impairment in social, occupational or other important areas of current functioning
> E. These disturbances are not better explained by intellectual disability (ID)†
>
> \*ICD-10-DCR also recognises the diagnosis of 'atypical autism', whereby the features of abnormal development only manifest for the first time after 3 years of age (A) and/or the affected individual does not satisfy all three of the clinical diagnostic criteria (B-D).
> †DSM-5 recognises that ASD and ID frequently coexist, but that "to make comorbid diagnoses of autism spectrum disorder and intellectual disability, social communication should be below that expected for general developmental level."

ASD appears to reduce the number of individuals satisfying the diagnostic criteria, with only about 50%–75% of individuals maintaining diagnoses, a phenomenon which disproportionately affects higher functioning non-intellectually disabled (non-ID) individuals.[6] However, the vast majority of what is known about ASD comes from research involving child populations, with less than 15% of UK-based ASD research being focused exclusively on adults,[7] with a similar focus on children seen internationally.[8]

When autism was first described in the medical literature, it was considered a rare condition.[9] However, there has been a significant rise in reported prevalence estimates in recent decades, which has been attributed to a myriad of factors, including broadening of diagnostic criteria, changes in study methodology, increasing knowledge of ASD among lay persons and professionals, and possibly a genuine increase in prevalence.[10]

ASDs represent a major global public health issue, responsible for over 111 disability adjusted life years per 100 000 persons.[11] Community-based prevalence rates for ASDs in adults within the general population show considerable variation across studies over time, although a recent large-scale epidemiological study estimated a prevalence rate of 1.1% (95% CI 0.3 to 1.9).[11 12]

A large UK-based study of 7274 participants in the 2014 Adult Psychiatric Morbidity Survey (APMS)[13] coupled with a register of 290 adults with intellectual disability (ID), defined by DSM-5 as deficits in intellectual functioning (typified by an IQ score of ≤70) coupled with deficits or impairments of adaptive functioning, with onset during the developmental period found an overall estimated ASD prevalence of 1.1% (95% CI 0.3% to 1.9%).[12] This is in keeping with the findings of a child study of 56 946 participants by Baird et al,[14] who found an overall estimated prevalence of ASDs of 116.1 per 10 000 (95% CI 90.4 to 141.8). However, while the Brugha et al[12] ASD prevalence findings for people with mild ID (1.0%, 95% CI 0.4 to 2.2) were similar to the overall findings, those with moderate to profound ID had a markedly increased risk of ASD, with prevalence rates of 42.3% (95% CI 31.1 to 54.3) and 35.2% (95% CI 23.5 to 49.0) in male and female, respectively.

However, relative to the community setting, there has been very little research into the prevalence of ASDs among adults within acute mental health inpatient settings.[15] Previous work appears to suggest that the prevalence of ASDs could be significantly greater in this group than that of the general population and that they may be substantially underdiagnosed in this group, with prevalence estimates varying from 1.5% to 9.9%.[16–19]

The vast majority of research on psychiatric comorbidity in patients with ASDs has been conducted in child and adolescent populations.[20] Such studies have shown high rates of comorbid psychiatric conditions among individuals with ASDs, with rates ranging from 70.8%[21] to 80.9%,[22] and commonly comorbid conditions including anxiety disorders, obsessive compulsive disorders, phobias, oppositional defiant disorder and attention deficit hyperactivity disorder.[21–25] Notably, a study by Brereton et al[25] found that children with ASDs experienced significantly higher levels of psychopathology compared with those with ID, a group which in itself is associated with an elevated risk of psychiatric disorder.[26]

However, it is unclear how the burden of psychiatric comorbidity seen in children with ASDs transitions into adult life.[27] It is essential to understand this natural course of ASDs and its comorbidities in order to develop appropriate services to meet the needs of this patient group.[28] Additionally, some psychiatric disorders, such as schizophrenia, do not typically manifest until late adolescence onwards,[29] so child comorbidity studies into such disorders would be of limited value.

A scoping review by Cashin et al[30] found a lack of research pertaining to the physical health of individuals with diagnosed ASDs, and that most studies were limited to retrospective analysis of healthcare records rather than involving any form of direct health assessment. Additionally, these studies were focused on individuals with previously recognised ASD diagnosed through routine clinical assessment, rather than having been identified through

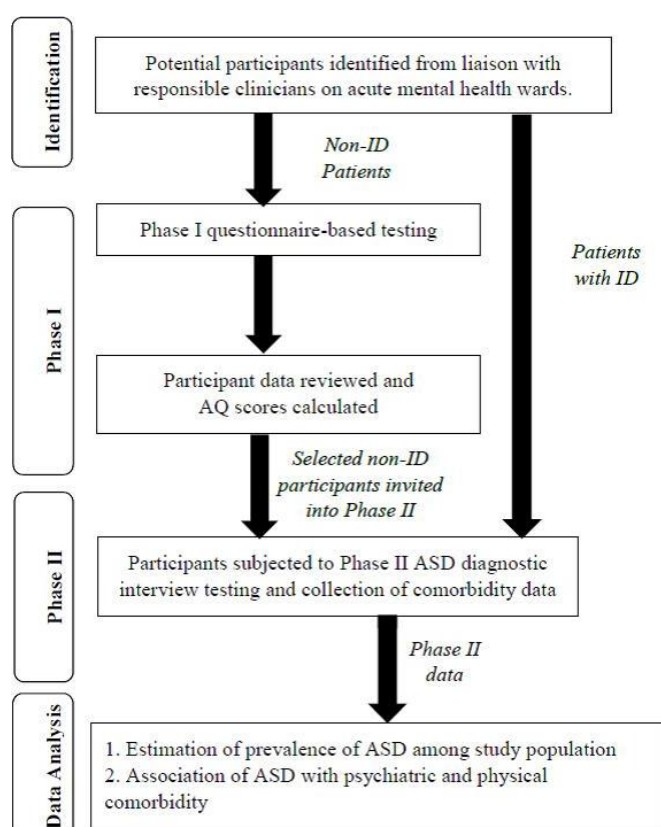

**Figure 1** Flow chart summary of study design. ASDs, Autism spectrum disorders; AQ, autism quotient

**Table 1** The probability of selection for Phase II according to total scores in the AQ questionnaire. This probability framework has been validated in a previous study conducted on adult participants from a mixture of inpatient and community mental health settings.[39] Participants scoring higher on these questionnaires will have a greater probability of selection

| AQ score | Probability of selection |
|---|---|
| ≤19 | 0.1 |
| 20–24 | 0.2 |
| 25–29 | 0.3 |
| 30–39 | 0.6 |
| ≥40 | 1.0 |

population-based epidemiological studies. Current evidence suggests that children with ASDs are at greater risk of a range of comorbid physical illness compared with their non-ASD peers, including obesity,[31–33] epilepsy,[34 35] asthma,[35] sleep disorders[36] and gastrointestinal disorders.[36] This assertion is further supported in a recent systematic review by Muskens *et al*,[37] who described medical disorders across a range of medical areas (including immunology, neurology and gastroenterology) as being widespread among children with ASD. However, despite the relative paucity of research in child populations, even less is known about the burden of physical comorbidity among adults with ASDs.[38]

The aims of this prospective pilot study are to estimate the prevalence of ASDs among adults based on acute mental health inpatient units, as well as to examine the association between ASDs and other psychiatric and physical comorbidities within this group.

## METHODS AND ANALYSIS
### Study design
The study (also known as the SPRINT study (The Prevalence of Social Communication PRoblems in Adult Psychiatric InpaTients)) will employ a multiple-phase cross-sectional design for participants without ID, and a single-phase design for participants with ID (figure 1). A multiple-phase approach to studying ASD prevalence has been used in several previous similar studies,[12 19 39] as the

ASD clinical interview-based diagnostic assessments are both time and resource intensive.[40] Thus, a multiphase approach permits coverage of a larger study population per unit of resource. However, such an approach is heavily reliant on the accuracy of the tool used in the first phase in identifying probable cases of ASD.

In terms of the multiple-phase design for non-ID patients, transition to Phase II will be selected via stratified random sampling according to the patient's scores on the Autism Quotient (AQ)[41] questionnaire (table 1). This weighted approach enables generation of a prevalence estimate for the population as a whole (compared with if there were a single threshold AQ score for Phase II progression) and has been extensively validated, including in the 2014 APMS.[13]

The rationale for a single-phase design in patients with ID is twofold. First, there is a lack of validated and relevant ASD questionnaires for this patient group.[42] Second, the prevalence of ASD in this patient group is substantially higher in community studies,[12] such that a single-phase design would be less resource-inefficient in identifying individuals meeting ASD diagnostic criteria within this group.

### Study population
The total sample will comprise 200 patients between the ages of 18 and 65 years, with a history of having been a psychiatric inpatient on an adult acute mental health ward during the study time period (from 6 March 2019 to 31 December 2020). The inclusion and exclusion criteria are summarised in table 2. Out of these 200 patients undergoing Phase I testing, 40 will transition to Phase II of the study. The Phase II patient group will consist of 25±5 non-ID patients, and 15±5 patients with ID.

Study participants will be recruited from two inpatient units based in Leicestershire, UK. These units serve a population of approximately 1 million people living in both urban and rural areas within Leicester, Leicestershire and Rutland.[43] Applications to extend the study to further sites are currently in progress.

**Table 2** Inclusion and exclusion criteria. Additionally, participants lacking capacity will be withdrawn from the study if they become distressed by the assessment process and/or their guardians are not in agreement with them remaining in the study

| Inclusion criteria | Exclusion criteria |
|---|---|
| Aged between 18 and 65 years. | <18 or >65 years of age. |
| Being or having been a psychiatric inpatient on an adult acute mental health ward during the study time period. | Having no history of being a psychiatric inpatient on an adult acute mental health ward during the study time period. |
| | Clinical diagnosis of dementia. |
| | Not understanding written and/or verbal English. |

## Patient and public involvement

The study design was refined following consultation group sessions with both patients with ID and their carers, as well as patients without ID and their carers. Significant changes made to the study design were made as a result of

these sessions, including reduction of the questionnaire and interview burden, as well as incorporation of a stigma measure for patients with ID, as this was considered a research priority by patients with ID and their carers.

Patients and carers will all receive a report summarising the findings of the study, unless they request not to, and an easy-read version of the report will be developed for patients with ID.

## Measures

The measures to be used are summarised in table 3. The Phase I measures, which will be used to test non-ID participants only, include the AQ,[41] which is recommended by the National Institute for Health and Care Excellence as referral tools for adults with possible ASD who do not have ID.[44] Also included in the test battery are ASD tools which are less comprehensively validated in adult populations, but appear to show promise, including the Informant version of the Social Responsiveness Scale, second edition[45] and the Self-report and Informant versions of the Adult Social Behaviour Questionnaire, ASBQ.[46] Phase I also includes a basic information form for participants,

**Table 3** Summary of study measures

| Measure | Purpose |
|---|---|
| **Phase I** (non-ID participants only) | |
| Autism Quotient, AQ[41] | Measuring likelihood of ASD. |
| Informant version of the Social Responsiveness Scale, second edition, SRS-A[45] | |
| Self-report and Informant versions of the Adult Social Behaviour Questionnaire, ASBQ[46] | |
| Self-report and Informant versions of the EuroQol-5D-5L, EQ-5D-5L[47] | Measuring quality of life. |
| Basic information form | Collecting information pertaining to patient demographics, as well as the 2014 Adult Psychiatric Morbidity Survey (APMS) mental illness and physical health conditions checklists.[13] |
| **Phase II** | |
| Diagnostic Interview for Social and Communication Disorders, DISCO[48] | Establishing whether participant meets diagnostic criteria for ASDs. |
| Autism Diagnostic Observation Schedule version 2, ADOS-2[49] | |
| The Mental Illness Diagnoses and Treatment section of the 2014 APMS[13] | Establishing participant's psychiatric and physical health comorbidities. |
| The Physical Health Conditions section of the 2014 APMS[13] | |
| ASD interview subsection of version 3 of the Schedules for Clinical Assessment in Neuropsychiatry, ASD-SCAN-3 | Field testing of interview schedule (for non-ID participants only). |
| Stigma Questionnaire for people with Intellectual Disability, SQID[50] | Participants experience of stigma (for ID participants only). |
| Basic information form | Collecting information pertaining to patient demographics, as well as the 2014 Adult Psychiatric Morbidity Survey (APMS) mental illness and physical health conditions checklists[13] (for ID participants only, as this form is completed in phase I for non-ID participants). |

ID, intellectual disability.

**Table 4** Summary of secondary outcome parameters

| Outcome category | Corresponding parameters | Study phase in which data is collected |
|---|---|---|
| Autism questionnaire data | ► Autism Quotient, AQ[41]<br>► Informant version of the Social Responsiveness Scale, second edition, SRS-A[45]<br>► Self-report and Informant versions of the Adult Social Behaviour Questionnaire, ASBQ[46] | ► Phase I |
| Basic demographic information | ► Date of birth<br>► Sex<br>► Postcode<br>► Ethnic group<br>► Employment status<br>► Relationship status | ► Phase I (non-ID participants)<br>► Phase II (ID participants) |
| Mental health history | ► Date of most recent psychiatric hospital admission<br>► Discharge date of most recent psychiatric hospital admission (where applicable)<br>► Total number of inpatient psychiatric admissions<br>► 2014 Adult Psychiatric Morbidity Survey (APMS) mental illness checklist[13] | ► Phase I (non-ID participants)<br>► Phase II (ID participants) |
| | ► The Mental Illness Diagnoses and Treatment section of the 2014 APMS[13] | ► Phase II |
| Physical health history | ► 2014 APMS physical health conditions checklist[13] | ► Phase I (non-ID participants)<br>► Phase 2 (ID participants) |
| | ► The Physical Health Conditions section of the 2014 APMS[13] | ► Phase II |
| General health | ► EuroQol-5D-5L[47] data for participants and informants | ► Phase I |
| Stigma experience (ID participants only) | ► Stigma Questionnaire for people with Intellectual Disability, SQID[50] | ► Phase II (ID participants) |

ID, intellectual disability.

including demographic details, encompassing basic demographic information and information pertaining to mental and physical health, including the 2014 APMS mental illness and physical health conditions checklists, thus enabling measurement harmonisation and wider comparison.[13] Additionally, both participants and informants will complete the EuroQol-5D-5L,[47] a self-report instrument that measures general health.

The Phase II measures include the DISCO[48] (although the interview will be limited to the diagnostic algorithm, psychiatric conditions and forensic problems items) and ADOS-2,[49] both of which are tools with established validity for the diagnosis of ASDs. Also, psychiatric and physical comorbidities will be assessed using the relevant sections of the 2014 APMS interview.[13] Other tests include the field testing of the ASD interview of version 3 of the Schedules for Clinical Assessment in Neuropsychiatry, as well as the Stigma Questionnaire for people with Intellectual Disability.[50]

### Outcome parameters

The primary outcome parameter for this pilot study is the presence or absence of meeting ICD-10-DCR and DSM-5 diagnostic criteria for ASDs, respectively. Secondary outcome parameters, including the study phase in

which the corresponding parameters are collected, are summarised in table 4.

### Data analysis plan

The primary outcome parameter will be based on patients who transition to Phase II of the study. The ASD prevalence estimate will be calculated based on the proportion of Phase II participants who meet the ICD-10-DCR and DSM-5 diagnostic criteria for ASD on clinical assessment, with adjustment for selection and non-response, as well as the different study designs for the ID and non-ID patient subgroups.

For the feasibility study, the analyses will be descriptive. For continuous variables, values and SD will be reported; for categorical variables, percentages and SD will be reported. We will use multiple imputation to account for any missing data. For the full-powered study, the generalised linear model will be used to estimate the difference between groups, as well as corresponding OR, 95% CI and p values.

The analyses will be primarily conducted in SPSS (version 26), but for more advanced statistical analyses (such as use of the generalised linear model) and graphical representation, we will use R.

## Sample size justification

The sample size was calculated with direct statistician support, as well as the use of a statistical textbook and related software.[51] For a significance level of 0.05 and power of 85%, assuming estimates of 1% community ASD prevalence and 5% inpatient ASD prevalence,[15] a sample size of 374 patients is required. Ideally, we would be subjecting a minimum of 374 patients to Phase II testing. This is because only participants completing Phase II interview assessments will yield data pertaining to the primary outcome variable (ie, the presence or absence of satisfying diagnostic criteria for ASD).

However, in the first instance, around 40 patients (from the minimum of 200 subjected to Phase I testing) will be selected for Phase II testing (via stratified random sampling for non-ID patients, whereas ID patients will progress automatically to Phase II). The Phase I sample size of 200 patients takes into account the probability sampling method for Phase II selection, in addition to allowing for losses to follow-up for Phase I participants invited for Phase II testing. Our results will assume that the data for these 40 patients are representative of the larger group subjected to Phase I testing. Owing to the sample size, the SPRINT study shou

ld be considered a pilot/feasibility study, the intention of which being to establish and further refine the technical, administrative and logistical aspects of the study, with a view towards sample size expansion on its completion. The sample size represents an appropriate sample size for a feasibility study of this type.[52 53]

## Ethics and dissemination

The written approval of the National Health Service Health Research Authority has been obtained, as well as the local ethics committee of the study centre. Written informed consent is and will be obtained from every participant. Significant modifications to the study protocol will be communicated to relevant members of the research team. Any results from this study will be published in peer-reviewed journals and conference proceedings. Additionally, the study's findings will be disseminated to all participants, including easy-read versions of the findings for those with IDs.

**Acknowledgements** The authors would like to acknowledge the following individuals for their contributions to the study: Nicola Spiers, Maria Viskaduraki and Zoe Morgan for statistical support. Elizabeta Mukaetova-Ladinska and Verity Chester for assistance with the systematic reviews underpinning the study. Dave Clarke and Henry Simkins for assistance with study design. Natalie Marking, Hannah Harrison, Tom Pringle, Deborah Glancy, Joanna McGarr, Rebekah Pole, Kris Roberts and Sam Pollen for support with study recruitment. Debra Bugler, Janice Holmes, Joy Fellows and Robin Oxley-Boyle for administrative support. Joanne Prosser and Naishali Chandarana for advice and support in developing easy-read materials. Freya Tyrer and Lesley Thoms for assistance with Patient and Public Involvement Sessions. The members of the Peoples Forum (Leicester, United Kingdom) and patients and carers at Leicestershire Partnership NHS Trust for providing valued input in the Patient and Public Involvement Sessions.

**Contributors** ST, TB, GLY, RK, RA and MA-U were involved in drafting the study protocol. TB and ST conceived of the idea for this study. ST is the principal investigator for the study, under the supervision of TB. ST led in writing the manuscript, TB and GLY contributed revisions and all authors approved the final manuscript.

**Funding** This study was supported by funding awarded by an NIHR Senior Investigator, as well as receiving funding and resource support from the NIHR, as a result of the study being adopted onto the NIHR Clinical Research Network Portfolio of studies. GLY's research is partly funded by an MRC grant.

**Competing interests** TB is directly involved with the development of the autism spectrum disorder (ASD) interview questions for the Schedules for Assessment in Neuropsychiatry version 3, which have not yet been formally published; this study is involved in testing the clinical utility of these questions. Additionally, TB and RK were authors of the 2014 APMS, from which two questionnaires are being used. Neither TB nor RK will be receiving any monetary payment as a result of the inclusion of the ASD interview questions or the 2014 APMS questionnaires within this study.

**Patient consent for publication** Not required.

**Provenance and peer review** Not commissioned; externally peer reviewed.

**ORCID iD**
Sam Tromans http://orcid.org/0000-0002-0783-285X

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
