## [Reviewer comments · BMJ Open]

ARTICLE DETAILS

TITLE (PROVISIONAL)	Study Protocol: An investigation of the prevalence of autism among adults admitted to acute mental health wards: A cross-sectional pilot study
AUTHORS	Tromans, Sam; Yao, Guiqing; Kiani, Reza; Alexander, Regi; Al-Uzri, Mohammed; Brugha, Traolach

VERSION 1 – REVIEW

REVIEWER	Guodong Liu Penn State University
REVIEW RETURNED	13-Sep-2019

GENERAL COMMENTS	This was a nicely written, short protocol paper. However, my major concern was the power of the study. The authors had carried a power calculation and clearly known their study won't promise any conclusive finding due to the lack of power. While they mentioned that this study should be considered as pilot study, and I must admit that it could end of being a good pilot/feasibility study, I just don't see the value of publishing a manuscript of a study protocol of a pilot study.
---

REVIEWER	Lauren Alexander Trinity College Dublin Ireland
REVIEW RETURNED	12-Oct-2019

GENERAL COMMENTS	Thank you for the opportunity to review this paper. This is an important piece of work. The study has clearly been thoroughly planned. I have just a few points that require clarification: 1. My understanding is that this will be a preliminary study, that will later be expanded to meet the calculated sample size of 374. However, I am unclear about the figures given for the present study. Is 200 the number of inpatients expected to be eligible for participation or is there some other rationale for this number? What will happen with the data for the 160 phase 1 participants that don't go on to phase 2? Will this be included in any analysis? Could the method illustrated in table 2 be explained in the text? It would be useful to state explicitly the purpose of stratified random sampling (rather than just selecting all participants above a cutoff point in the AQ).2. There is some data in the abstract about the breakdown of ID and non-ID participants that is not in the main text.3. Primary outcome is well defined. Secondary outcomes should be listed clearly, rather than "secondary outcome parameters include...." Is the results of the stigma questionnaire planned to be part of the secondary outcomes?
--

	4. The plan for data analysis is a little vague. Which tests specifically will be used? Why do they say that either SPSS or R will be used? Will they each be used for different parts of the analysis or has this not yet been decided? Please clarify in the text. 5. A couple of grammatical errors... whom used instead of who (3 instances) On page 11, the word 'required' is repeated.
--	---

VERSION 1 – AUTHOR RESPONSE

COMMENTS TO THE AUTHORS:	RESPONSE	LOCATION IN MANUSCRIPT AND DETAILS OF CHANGES
Reviewer #1:		
REVIEW 1 COMMENT 1: This was a nicely written, short protocol paper. However, my major concern was the power of the study. The authors had carried a power calculation and clearly known their study won't promise any conclusive finding due to the lack of power. While they mentioned that this study should be considered as pilot study, and I must admit that it could end of being a good pilot/feasibility study, I just don't see the value of publishing a manuscript of a study protocol of a pilot study.	We thank the reviewer for their feedback on our manuscript, and note their concern about our study being a pilot study. However, we are conducting this study with a view to establishing and refining the technical, administrative and logistic feasibility of a full-scale study, including testing the acceptability, recruitment and intervention package; all of which will provide valuable information for future research. Publishing the protocol of the pilot study aims to improve the transparency in conducting and report of the full trial (https://pilotfeasibilitystudies.biomedcentral.com/articles/10.1186/s40814-019-0499-1). Additionally, BMJ Open have a track record of publishing protocols for pilot studies and include specific guidance pertaining to submitting such studies on their author guidelines.	(Sample size justification, third paragraph) Owing to the sample size, the SPRINT study should be considered a pilot/feasibility study, the intention of which being to establish and further refine the technical, administrative and logistical aspects of the study, with a view towards sample size expansion upon its completion.
Reviewer #2		
REVIEWER 2 COMMENT 1: Thank you for the opportunity to review this paper. This is an important piece of work. The study has clearly been thoroughly planned. I have just a few points that require clarification: 1. My understanding is that this will be a preliminary study, that will later be expanded to meet the calculated sample size of 374. However, I am unclear about the figures given for the present study.	We thank the reviewer for their kind comments about the manuscript. We agree that the sample size required some further justification, and we have endeavoured to do this, citing relevant literature.	(Sample size justification, second paragraph) The Phase 1 sample size takes into account the probability sampling method for Phase 2 selection, in addition to allowing for losses to follow-up for Phase 1 participants invited for Phase 2 testing. (Sample size justification, second paragraph) The sample size represents an appropriate sample size for a feasibility study of this type. ^{51,52}

Is 200 the number of inpatients expected to be eligible for participation or is there some other rationale for this number?		(References) 51. Johanson GA, Brookes, GP. Initial scale development: sample size for pilot studies. Educ Psychol Meas . 2010;70:394-400. 52. Hertzog MA. Considerations in determining sample size for pilot studies. Res Nurs Health. 2008;31:180-191.
REVIEWER 2 COMMENT 2: What will happen with the data for the 160 phase 1 participants that don't go on to phase 2? Will this be included in any analysis?	Important secondary outcome data will be obtained for all participants, including those whom are not selected for Phase 2. We have outlined details pertaining to a basic information form for phase 1 patients (and phase 2 patients with ID), as well as included details in a new table on secondary outcome parameters (Table 5), which includes details relating to the Phase in which particular data are collected.	(Measures, first paragraph) Phase 1 also includes a basic information form for participants, including demographic details, encompassing basic demographic information, and information pertaining to mental and physical health, including the 2014 APMS mental illness and physical health conditions checklists, thus enabling measurement harmonisation and wider comparison.¹³ (Table 4 – Basic information form incorporated into table). (Outcome parameters, first paragraph) Secondary outcome parameters, including the study phase in which the corresponding parameters are collected, are summarised in Table 5. (Table 5 – New Table – Outlines all secondary outcome parameters as well as the corresponding study phase in which said data is collected).
REVIEWER 2 COMMENT 3: Could the method illustrated in table 2 be explained in the text? It would be useful to state explicitly the purpose of stratified random sampling (rather than just selecting all participants above a cutoff point in the AQ).	Thank you for identifying this issue, we have provided details within the manuscript justifying the rationale for the sampling approach.	(Study design, second paragraph) This weighted approach enables generation of a prevalence estimate for the population as a whole (compared to if there were a single threshold AQ score for Phase 2 progression), and has been extensively

		validated, including in the 2014 APMS. ¹³
REVIEWER 2 COMMENT 4: There is some data in the abstract about the breakdown of ID and non-ID participants that is not in the main text.	Thank you for pointing this out, we agree that this information should also be included in the main body of the manuscript.	(Study population, first paragraph) The Phase 2 patient group will consist of 25±5 non-ID patients, and 15±5 patients with ID.
REVIEWER 2 COMMENT 5: Primary outcome is well defined. Secondary outcomes should be listed clearly, rather than "secondary outcome parameters include...." Is the results of the stigma questionnaire planned to be part of the secondary outcomes?	We are grateful to the reviewer for mentioning this, and do agree that the secondary outcome measures could be made more clear to the reader. We have now added a table listing the secondary outcome measures, of which the results of the stigma questionnaire are one.	(Outcome parameters, first paragraph) Secondary outcome parameters, including the study phase in which the corresponding parameters are collected s, are summarised in Table 5. (Additional table – Table 5)
REVIEWER 2 COMMENT 6: The plan for data analysis is a little vague. Which tests specifically will be used? Why do they say that either SPSS or R will be used? Will they each be used for different parts of the analysis or has this not yet been decided? Please clarify in the text.	We appreciate that this was ambiguous. We have provided further details pertaining to the data analysis plan.	(Data analysis plan, first, second and third paragraphs) The primary outcome parameter will be based on patients who transition to Phase 2 of the study. The ASD prevalence estimate will be calculated based on the proportion of Phase 2 participants who meet the ICD-10-DCR and DSM-5 diagnostic criteria for ASD on clinical assessment, with adjustment for selection and non-response, as well as the different study designs for the ID and non-ID patient subgroups. For the feasibility study, the analyses will be descriptive. For continuous variables, values and standard deviations will be reported; for categorical variables, percentages and standard deviations will be reported. We will use multiple imputation to account for any missing data. For the full-powered study, the generalised linear model will be used to estimate the difference between groups, as well as corresponding OR, 95% CI and p-values. The analyses will be primarily conducted in SPSS, but for more advanced statistical analyses (such as use of the

		generalised linear model) and graphical representation, we will use R.
REVIEWER 2 COMMENT 7: A couple of grammatical errors... whom used instead of who (3 instances) On page 11, the word 'required' is repeated.	We thank the reviewer for identifying these grammatical errors, which we have changed.	(Introduction, fifth paragraph) 'Whom' changed to 'who.' (Introduction, ninth paragraph) 'Whom' changed to 'who.' (Measures, first paragraph) 'Whom' changed to 'who.' (Sample size justification, first paragraph) One instance of the word 'required' has been removed.

VERSION 2 – REVIEW

REVIEWER	Lauren Alexander Trinity College Dublin Ireland
REVIEW RETURNED	25-Nov-2019
GENERAL COMMENTS	None.